# Correlation between B7-H4 and Survival of Non-Small-Cell Lung Cancer Patients Treated with Nivolumab

**DOI:** 10.3390/jcm8101566

**Published:** 2019-10-01

**Authors:** Carlo Genova, Simona Boccardo, Marco Mora, Erika Rijavec, Federica Biello, Giovanni Rossi, Marco Tagliamento, Maria Giovanna Dal Bello, Simona Coco, Angela Alama, Irene Vanni, Giulia Barletta, Rita Bianchi, Claudia Maggioni, Paolo Bruzzi, Francesco Grossi

**Affiliations:** 1Lung Cancer Unit, IRCCS Ospedale Policlinico San Martino, Largo Rosanna Benzi 10, 16132 Genoa, Italy; simona.boccardo@hsanmartino.it (S.B.); giovanni.rossi.1689@gmail.com (G.R.); tagliamento.marco@gmail.com (M.T.); mariagiovanna.dalbello@hsanmartino.it (M.G.D.B.); simona.coco@hsanmartino.it (S.C.); angela.alama@hsanmartino.it (A.A.); irenevanni85@yahoo.it (I.V.); giulia.barletta@yahoo.it (G.B.); claudia_87m@yahoo.it (C.M.); 2Pathology Unit, IRCCS Ospedale Policlinico San Martino, Largo Rosanna Benzi 10, 16132 Genoa, Italy; marco.mora@hsanmartino.it (M.M.); bianchirita22@gmail.com (R.B.); 3Medical Oncology Unit, Fondazione IRCCS Ca’ Granda Ospedale Maggiore Policlinico, Via Francesco Sforza, 28, 20122 Milan, Italy; ery80x@yahoo.it (E.R.); francesco.grossi@policlinico.mi.it (F.G.); 4Oncology Unit, Ospedale Maggiore della Carità, Corso Mazzini 18, 28100 Novara, Italy; febiello@gmail.com; 5Department of Epidemiology, Biostatistics and Clinical Trials, IRCCS Ospedale Policlinico San Martino, Largo Rosanna Benzi 10, 16132 Genoa, Italy; paolo_bruzzi@libero.it

**Keywords:** non-small-cell lung cancer, nivolumab, B7-H4, immune checkpoint inhibitors, immunohistochemistry

## Abstract

Reliable predictors of benefit from immune checkpoint inhibitors in non-small-cell lung cancer (NSCLC) are still limited. We aimed to evaluate the association between the expression of selected molecules involved in immune response and clinical outcomes in NSCLC patients receiving nivolumab. In our study, the outcomes of 46 NSCLC patients treated with nivolumab in second or subsequent lines (Nivolumab Cohort) were compared with the expression of PD-L1, PD-L2, PD-1, B7-H3, and B7-H4 assessed by immunohistochemistry (IHC). Samples from 17 patients (37.0%) in the Nivolumab Cohort were positive for B7-H4 expression. At univariate analyses, only B7-H4 expression was associated with significantly decreased progression-free survival (PFS; 1.7 vs. 2.0 months; *p* = 0.026) and with a disadvantage in terms of overall survival (OS) close to statistical significance (4.4 vs. 9.8 months; *p* = 0.064). At multivariate analyses, B7-H4 expression was significantly associated with decreased PFS (hazard ratio (HR) = 2.28; *p* = 0.021) and OS (HR = 2.38; *p* = 0.022). Subsequently, B7-H4 expression was compared with clinical outcomes of 27 NSCLC patients receiving platinum-based chemotherapy (Chemotherapy Cohort), but no significant association was observed. Our results suggest a negative predictive role of B7-H4 in a population of NSCLC treated with immune checkpoint inhibitors, which deserves further research.

## 1. Introduction

The treatment of advanced non-small-cell lung cancer (NSCLC) has been revolutionized by the introduction of immune checkpoint inhibitors in clinical practice. In particular, the compounds directed against the programmed death protein 1 (PD-1) on T-cell surface or its ligand (PD-L1) on tumor cells have significantly improved overall survival (OS) of previously treated NSCLC patients compared to chemotherapy; similarly, treatment-naïve NSCLC patients with strong expression of PD-L1 achieved longer OS with PD-1 blockade compared to first-line chemotherapy [1,2]. The expression of PD-L1, evaluated by immunohistochemistry (IHC) in formalin-fixed paraffin-embedded (FFPE) tissue samples is currently considered a biomarker for checkpoint inhibitors targeting the PD-1/PD-L1 axis. Furthermore, different diagnostic tests for PD-L1 have been developed and validated; hence, many efforts to standardize the PD-L1 assessment have been performed [3,4].

However, the expression of PD-L1 alone has a limited role in selecting whether patients should receive an immune checkpoint inhibitor or chemotherapy in second or subsequent lines of treatment. In first place, it has been observed that different assays yield discordant results; additionally, PD-L1 has been recognized to have a heterogeneous expression within the same tumor, associated with variable and not completely understood immune mechanisms; third, and most important, a beneficial effect of anti-PD1 compounds was seen, in most studies, also in patients without detectable PD-L1 expression [5,6,7,8]. Thus, identifying other potential biomarkers might improve patient selection for treatment with immune checkpoint inhibitors.

According to preclinical studies, the cross-talk during immune response to cancer is not limited to the PD-1/PD-L1 axis and involves different molecules, either acting as co-stimulators or as immune checkpoints. For instance, PD-L2 binds PD-1 with inhibitory function, similarly to PD-L1; however, while PD-L1 is expressed in many cell lines, PD-L2 has a more restricted expression pattern, limited to dendritic cells, macrophages, and mast cells. Furthermore, PD-L2 might be associated with immune tolerance to normal respiratory cells [9,10]. Other molecules, such as B7-H3 and B7-H4, have been acknowledged as potential regulators of immunity and as prognostic factors in solid tumors [11]. B7-H3 has a controversial role in T-cell response [12,13,14,15] and its expression on tumor cells is reportedly associated with poor prognosis in NSCLC [16,17]. B7-H4 is a trans-membrane protein suggested to inhibit the activation and the clonal expansion of CD4^+^ and CD8^+^ cells, as well as the production of immune-promoting cytokines; notably, the receptor of B7-H4 on immune cells has not been identified yet [18,19]. In literature, high expression of B7-H4 has been associated with poor prognosis in different solid tumors [20,21,22,23]. Notably, co-expression of B7-H3 with PD-L1 and B7-H4 appears to be relatively uncommon, supporting the concept of a nonredundant biological role of these molecules, which might enact parallel mechanisms of immune escape [24].

While all the aforementioned biomarkers seem to have a role in the modulation of immune response and prognostic significance in NSCLC, their role in predicting response to immune checkpoint inhibitors has yet to be clarified. Since these potential biomarkers appear to play a role in tumor escape from immune system, our hypothesis was that their expression could lead to maintained ability to evade immune response in spite of PD-1 blockade and ultimately reduced clinical efficacy of PD-1 immune checkpoint inhibitors. Eventually, the identification of clinically relevant immune checkpoints might lead to the development of novel inhibitors, administered alone or in combination with PD-1/PD-L1 blocking agents in the future.

The aim of this study was to assess the potential correlations between the intratumor expression of a panel of immune-related biomarkers (PD-L2, PD-1, B7-H3, and B7-H4) and the clinical outcomes of advanced NSCLC patients treated with nivolumab for advanced NSCLC (Nivolumab Cohort) in order to determine the possible role of such molecules as potential markers of clinical efficacy of immune checkpoint blockade. Additionally, in order to define whether any meaningful biomarker identified in the Nivolumab Cohort might have a role in predicting the efficacy of immunotherapy, rather than a plain prognostic role, we retrospectively assessed the correlations between such biomarkers and the outcomes of a population of patients who had been treated with platinum-based chemotherapy and had not subsequently received any immune checkpoint inhibitor (Chemotherapy Cohort).

## 2. Experimental Section

### 2.1. Nivolumab Cohort

The Nivolumab Cohort included 46 patients affected by advanced NSCLC treated within the Italian Nivolumab Expanded Access Program (NCT02475382) and enrolled in a mono-institutional translational research study approved by our local ethics committee (registry number: P.R. 191REG2015) [25,26]. The patients were eligible if they met the following criteria: (i) cytologically or histologically confirmed advanced/metastatic NSCLC, (ii) progression after at least one line of platinum-based chemotherapy, (iii) Eastern Cooperative Oncology Group Performance Status (ECOG-PS) = 0–2, (iv) no previous treatment with immune checkpoint inhibitors, (iv) any brain metastasis had to be treated and clinically stable for at least 14 days before starting nivolumab, (v) no treatment with corticosteroids at a dose higher than 10 mg/day of prednisone or equivalent. Eligible patients received nivolumab at 3 mg/kg every 14 days, with assessment by computed tomography scan (CT-scan) every 8 weeks. Nivolumab was administered until onset of unacceptable toxicities, patient’s refusal, death, or up to 96 weeks from the start of treatment; treatment beyond tumor progression was allowed based on investigators’ judgment as long as clinical benefit was perceived. Objective responses and progression-free survival (PFS) were determined according to the Response Evaluation Criteria in Solid Tumors (RECIST) v.1.1; due to the peculiar mechanism of action of nivolumab, we assessed objective responses also with immune-related response criteria (irRC). Progression-free survival (PFS) and OS were calculated from the first administration of nivolumab to progression/death. 

### 2.2. Chemotherapy Cohort

The Chemotherapy Cohort included 27 treatment-naive patients with histologically or cytologically confirmed advanced non-squamous NSCLC and ECOG PS = 0–1 treated between 2011 and 2015 and drawn from a wider population of patients (*n* = 90) enrolled in a mono-institutional translational research study (NCT02055144) on the basis of available stored tissue for biomarker analyses. The regimen of choice was cisplatin plus pemetrexed for up to 4 cycles, followed by maintenance with pemetrexed. Carboplatin was administered in place of cisplatin to patients with a creatinine clearance < 60 mL/min. Tumor response was assessed with RECIST v.1.1 every 2 cycles. Chemotherapy was administered until unacceptable toxicity, patient’s refusal, progression, or death. The aforementioned translational research study admitted patients with both squamous and non-squamous histology; however, to date, only the population affected by non-squamous histology completed accrual and was, hence, available for this analysis [27,28]. 

### 2.3. Immunohistochemistry (IHC)

Sections of FFPE tissue were cut at 2 µm and mounted on positively charged, adhesive glass slides (Superfrost Plus Gold, Thermo Scientific, Braunschweig, Germany). The IHC was carried out manually, miming the automated staining steps performed by Dako Autostainer Link 48, as indicated by the approved FDA protocol (PMA P150025; validated automated assay with rabbit monoclonal anti-human PD-L1 antibody clone 28–8 Pharm DX Dako-cat. n° SK005). The sections were heated at 60 °C for 15 min and washed with xylene (2 × 10 min) and ethanol (2 × 10 min) to remove paraffin. Antigen retrieval and primary antibody incubation were carried out at room temperature in a hydrate chamber. Two sections of human placenta were included in each run as control; one section was incubated with the primary antibody containing the PD-L1 rabbit monoclonal antibody, while a second section was incubated with the negative control reagent of the kit, an IgG rabbit monoclonal antibody in a buffer solution. A two-step immunoperoxidase staining method was used for all the antibodies (EnVision+Dual Link System–HRP–DAB+, Dako-cat. n° K4065) as follows: B7-H4 (mouse monoclonal, clone MIH43, Abcam-cat. n° ab110221-dilution 1:60), B7-H3/CD276 (rabbit polyclonal, NovusBio–cat. n° NBP1-88965-dilution 1:25), PD-L2 (mouse monoclonal, clone 8G8, LSBio-cat. n° ab200377-dilution 1:50), and PD-1 (rabbit polyclonal, Abcam-cat. n° ab92484-dilution 1:50). Each run contained a positive control (on-slide tonsil tissue for PD-1 and PD-L2, prostate cancer for B7-H3, breast carcinoma for B7-H4) and a negative control (no primary antibody). The 22C3 antibody (mouse monoclonal anti-human PD-L1 antibody clone 28–8 Pharm DX Dako-cat. n° SK006) was obtained from the commercially available PD-L1 PharmDX kit on the BenchmarkULTRA (Ventana Medical Systems/Roche) platform, using the UltraView detection kit (UltraViewUniversal DAB detection Kit, Ventana Medical Systems/Roche–cat. n° 760–500) [29]. The percentage of stained positive tumor cells was evaluated for each sample under light microscope by two independent pathologists who were unaware of patient outcome. B7-H4, PD-1, and PD-L2 staining was observed in cytoplasm, while B7-H3 was detected in both cell membrane and cytoplasm and PD-L1 was exclusively expressed in cell membrane. Positive staining was defined as complete and/or partial circumferential membrane staining and/or diffuse cytoplasmic staining at any intensity based on the marker expression.

### 2.4. IHC Scoring

The expression of all the biomarkers (in the two cohorts) was evaluated by a score based on the number of stained tumor cells with minimum of 100 evaluable cells. The percentage of staining was arbitrarily graded as follows: <1% (negative), 1%–9% (low expression), 10%–49% (moderate expression), ≥50% (high expression) regardless of any intensity. The evaluable samples were then dived in negative (<1%) and positive (≥1% tumor cells) as previously described by Philips and colleagues [30].

### 2.5. Statistical Analysis

In the Nivolumab Cohort, response categories according to RECIST v.1.1 and irRC were compared with the expression of each biomarker under study with Fisher’s test or Chi-square test, as appropriate. Survival curves were compared between patient subgroups based on each biomarker expression by using Kaplan–Meier estimator. Cox’s proportional hazard model was used for multivariate survival analyses, starting from a model that included clinical and pathological characteristics, as well as the expression of the immune-related biomarkers. The clinical pathological features employed in the multivariate analyses for both cohorts included age (cut-off for elderly patients: 70 years), gender, smoking habit, and ECOG PS (0 vs. ≥1); additionally, histology and previous lines of treatment (1–2 vs. ≥3) were included in the multivariate analyses for the Nivolumab Cohort. The final model was reached by means of a stepwise regression with backward elimination of variables not significantly associated with PFS or OS, respectively, based on the likelihood ratio test. The same analyses were then repeated in the Chemotherapy Cohort, the rationale being that any difference between the 2 cohorts in the prognostic role of a specific biomarker would suggest an association between the expression of said biomarker and the efficacy of immunotherapy. Due to the exploratory aims of these analyses, only odds ratio for objective response and hazard ratios for PFS and OS for each of the 5 biomarkers in the 2 cohorts are reported, with their 95% CI, and no formal statistical comparison was planned. The analyses were carried out by using SPSS (v.23.0.0.0) and XLSTAT (v.19.03.44845).

## 3. Results

### 3.1. The Expression of B7-H4 Was Associated with Lower Progression-Free Survival in the Nivolumab Cohort

The Nivolumab Cohort included 46 evaluable patients whose clinical characteristics are summarized in Table 1. PD-L1 and PD-1 were not evaluable in one and two samples, respectively, due to excessive background. Representative images of IHC staining of PDL-1 and B7-H4 are reported in Figure 1, while the expression of each immune-related parameter is reported in Table 2 and in Appendix A. All the biomarkers, apart from PD-1, showed an expression <1% in most samples; consistent with the cut-off values of previous studies involving immune checkpoint inhibitors in pretreated NSCLC patients [31,32,33], we selected the value of 1% as an appropriate cut-off for defining positive (≥1%) vs. negative (<1%) samples for each potential biomarker. No significant correlations were observed among the expressions of the biomarkers.

When clinical characteristics were compared with each biomarker, PD-L2 expression was associated with ECOG PS = 0 (*p*-value = 0.038), while PD-1 expression was associated with age ≥70 years (*p-*value = 0.026) and B7-H3 expression was associated with squamous histology (*p-*value = 0.023). No statistically significant association between the expression of the immune-related biomarkers and the proportion of objective response rate (ORR) and disease control rate (DCR) was observed, although, notably, none of the six patients expressing B7-H3 achieved objective response Data involving ORR and DCR based on biomarkers’ expression are reported in Appendix A.

Progression-free survival was evaluable for 44 out of 46 patients according to RECIST. One patient discontinued treatment before the first assessment and did not undergo further CT scans after baseline; another patient had non-measurable disease according to RECIST, but was considered evaluable by irRC, as the baseline lesions met the requirements for measurability with such criteria. Seven patients died before undergoing the first response evaluation. The overall outcome data are reported in Appendix A. With regards to univariate analysis, we did not observe any association between RECIST-PFS and the expression of PD-L1 (Figure 2), PD-L2 (Figure 3), and PD-1 (Figure 4). B7-H3 expression was associated with significantly lower RECIST-PFS (median PFS 1.6 vs. 2.0 months; *p-*value = 0.009, Figure 5); while this result should be considered with caution due to the fact that only six out of 44 patients expressed B7-H3, the rapid progression of all these patients was noteworthy. More importantly, B7-H4 expression was associated with significantly reduced RECIST-PFS (median 1.7 vs. 2.0 months; *p*-value = 0.026, Figure 6). The irRC-PFS analyses based on each immune-related biomarker were generally consistent with RECIST-PFS analyses (Appendix A), although the irRC-PFS difference based on B7-H3 expression fell short of statistical significance (*p-*value = 0.057). In the multivariate RECIST-PFS analysis, the only variables significantly associated with shorter PFS were B7-H3 expression (HR = 4.14; 95% CI: 1.44–11.9; *p-*value = 0.019) and B7-H4 expression (HR = 2.28; 95% CI = 1.16–4.48; *p-*value = 0.021). 

In the univariate OS analyses, no significant association with immune-related biomarkers was observed (Figure 2, Figure 3, Figure 4, Figure 5 and Figure 6), although the association between B7-H4 expression and reduced OS was close to significance (4.37 vs. 9.83 months; *p*-value = 0.064, Figure 6). In multivariate analysis, OS was significantly reduced in patients with ECOG-PS = 1–2 vs. PS = 0, (HR = 2.73; 95% CI = 1.21–6.15; *p-*value = 0.01) and in patients with B7-H4 expression (HR = 2.38; 95% CI = 1.16–4.91; *p-*value = 0.022). A weak, nonsignificant association between PD-L1 expression and OS was observed (HR = 0.60; 95% CI = 0.15 -2.37 *p-*value = 0.460). 

### 3.2. B7-H4 Expression Was Not Associated with Response or Survival in the Chemotherapy Cohort

Tumor samples from 27 NSCLC patients within the Chemotherapy Cohort were collected. Since the available tissue was limited, we focused this analysis on B7-H4 and PD-L1 on the basis of the results concerning OS observed in the Nivolumab Cohort. The patients’ characteristics are summarized in Table 1, while the expressions of PD-L1 and B7-H4 are reported in Table 2 and in Appendix A. We applied the same cut-off for positivity already employed in the Nivolumab Cohort (≥1% vs. <1%). No correlation was observed between PD-L1 and B7-H4 expression. PD-L1 positivity was not associated with any clinical feature, while B7-H4 expression was associated with female gender (*p*-value = 0.008). The global outcome data for the Chemotherapy Cohort are reported in Appendix A. No significant correlation was observed between the expression of any biomarker and ORR or DCR, as reported in Appendix A. No association between biomarker expression and survival was observed: in particular, at the univariate analysis, PD-L1 expression did not significantly affect either RECIST-PFS (4.8 vs. 3.3 months; *p-*value = 0.444), or OS (12.8 vs. 7.4 months; *p-*value = 0.406), as reported in Figure 7; similarly, patients with B7-H4 expression ≥1% vs. <1% had similar median RECIST-PFS (3.3 vs. 3.4 months; *p-*value = 0.274) and OS (8.7 vs. 8.2 months; *p-*value = 0.284), as reported in Figure 8. At the multivariate analysis, RECIST-PFS was not associated with B7-H4 expression (HR = 0.64; 95% CI = 0.29–1.44; *p-*value = 0.275) or PD-L1 expression (HR = 0.74; 95% CI = 0.29–1.90; *p-*value = 0.446); likewise, OS was not associated with B7-H4 expression (HR = 0.85; 95% CI = 0.36–2.03; *p-*value = 0.287) or PD-L1 expression (HR = 0.50; 95% CI = 0.18–1.38; *p*-value = 0.408).

## 4. Discussion

While immune checkpoint blockade is now a consolidated approach in the management of advanced NSCLC, the need for reproducible predictive biomarkers with adequate ability to discriminate between those patients who will and those who will not benefit from immunotherapy has been repeatedly stressed. Our study focused on the expression assessment of a panel of potential immune-related biomarkers with the aim of identifying possible correlations with the outcomes of patients receiving nivolumab for advanced NSCLC. In the Nivolumab Cohort, the expression of B7-H4 was significantly correlated with a reduced PFS and also with a shorter OS, although this association was only close to statistical significance at the time of our analysis. Notably, at a median follow-up time of 18 months, ten patients out of 29 who had B7-H4 expression <1% were still alive as compared to only two patients out of 12 in the B7-H4-positive group. Conversely, B7-H4 expression was not associated with shorter RECIST-PFS or OS in the Chemotherapy Cohort, suggesting that B7-H4 expression might be associated with worse prognosis only in patients receiving immunotherapy. Despite the fact that the clinical meaning of B7-H4 in NSCLC has been debated for many years, a recent meta-analysis comprising nine studies, for a total of 1444 patients with NSCLC at any stage, demonstrated a correlation between B7-H4 expression and clinicopathological features such as poor differentiation, advanced disease stage, and poor survival, suggesting a prognostic role of this biomarker [21]. However, our results were not confirmed when B7-H4 expression was assessed in the cohort of patients receiving chemotherapy either for PFS or OS, suggesting that the role of B7-H4 in advanced NSCLC might depend on the administered treatment, with particular reference to immunotherapy. We acknowledge that the two populations of our study differ, as the ‘Nivolumab Cohort’ included pretreated patients with both histologic sub-types (squamous and non-squamous NSCLC), while the ‘Chemotherapy Cohort’ included only patients with adenocarcinoma receiving first-line chemotherapy. Although the composition of the cohorts could limit any formal statistical comparison, the clearly different behavior of B7-H4 in the two groups supports the potential negative predictive role of this biomarker for patients receiving nivolumab. 

Interestingly, PD-L1 and B7-H4 showed an opposite effect in the Nivolumab Cohort, although only the effect of B7-H4 was statistically significant. They are both known to promote immune tolerance; more specifically, while the immuno-regulatory role of PD-1/PD-L1 axis is widely recognized [34], B7-H4 likely inhibits the proliferation of T-effectors [35,36] and favors the proliferation of regulatory T-cells with inhibitory functions [37]. On the basis of our findings, we might speculate that, while PD-L1 expression is expected to improve the effect of PD-1/PD-L1 inhibitors, B7-H4 expression might promote immune tolerance favoring tumor escape in spite of PD-1/PD-L1 blockade. With regards to B7-H3 expression, despite the interesting report of decreased PFS within our Nivolumab Cohort, in line with other publications [38], the number of B7-H3-positive patients was too small to allow any conclusion on its potential role.

We are aware that our study has several limitations, mostly deriving from its nature of being a mono-institutional, retrospective study based on two different cohorts of patients affected by advanced NSCLC. Firstly, the number of patients included in each cohort was limited due to the availability of tissue samples considered suitable for our analysis. Secondly, there were meaningful differences in the cohorts, resulting in a limitation to the available aggregate analyses; however, the mono-institutional approach ensured that all the clinical assessments, including PFS and OS data collection, as well as IHC and analysis, were performed consistently among all the patients.

## 5. Conclusions

To the best of our knowledge, this is the first clinical study reporting a potential role of B7-H4 expression as predictor of benefit from PD-1 blockade with nivolumab in NSCLC; in spite of the reported limitations, our findings strongly encourage future prospective studies designed to define and eventually confirm the predictive role of B7-H4 expression for patients receiving immune checkpoint blockade for advanced NSCLC.

## Figures and Tables

**Figure 1 jcm-08-01566-f001:**
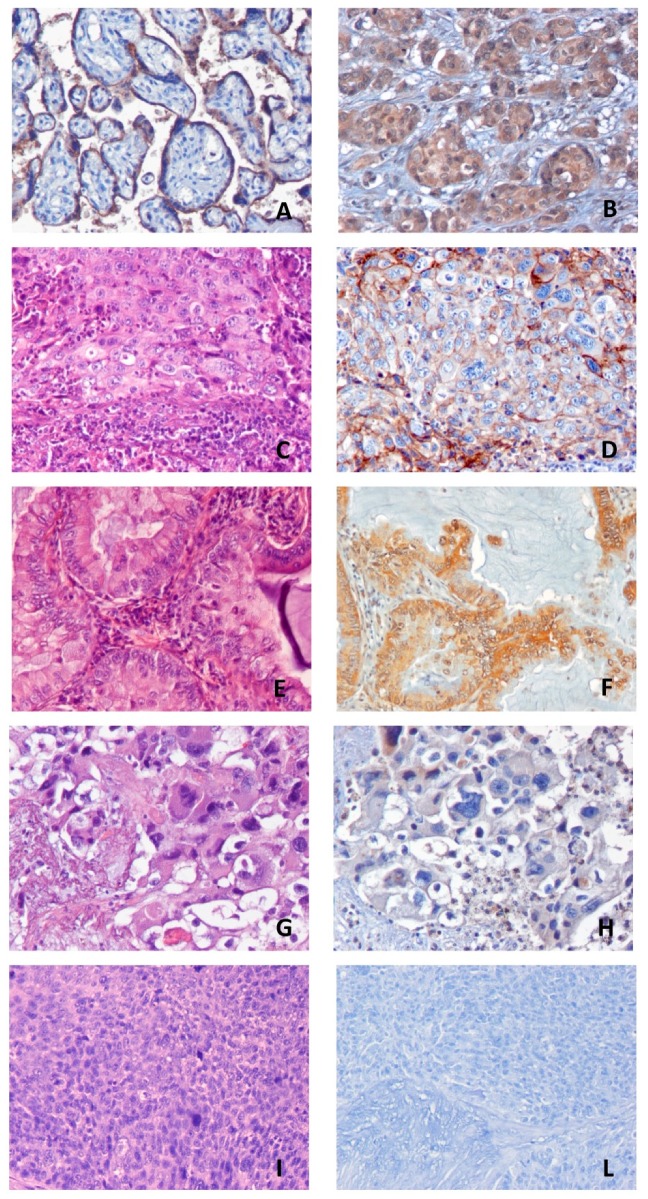
Representative images of immunohistochemistry (IHC) positive controls (**A**,**B**) and tumor samples from the Nivolumab Cohort (**C**–**L**). (**A**): positive PD-L1 control (placenta); (**B**): positive B7-H4 control (breast carcinoma); (**C**,**D**): Hematoxylin and eosin (H&E) and IHC positive staining of PD-L1; (**E**,**F**): H&E and IHC positive staining of B7-H4; (**G**,**H**): H&E and IHC negative staining of PD-L1; (**I**,**L**): H&E and IHC negative staining of B7-H4. All the images are at magnification 20×.

**Figure 2 jcm-08-01566-f002:**
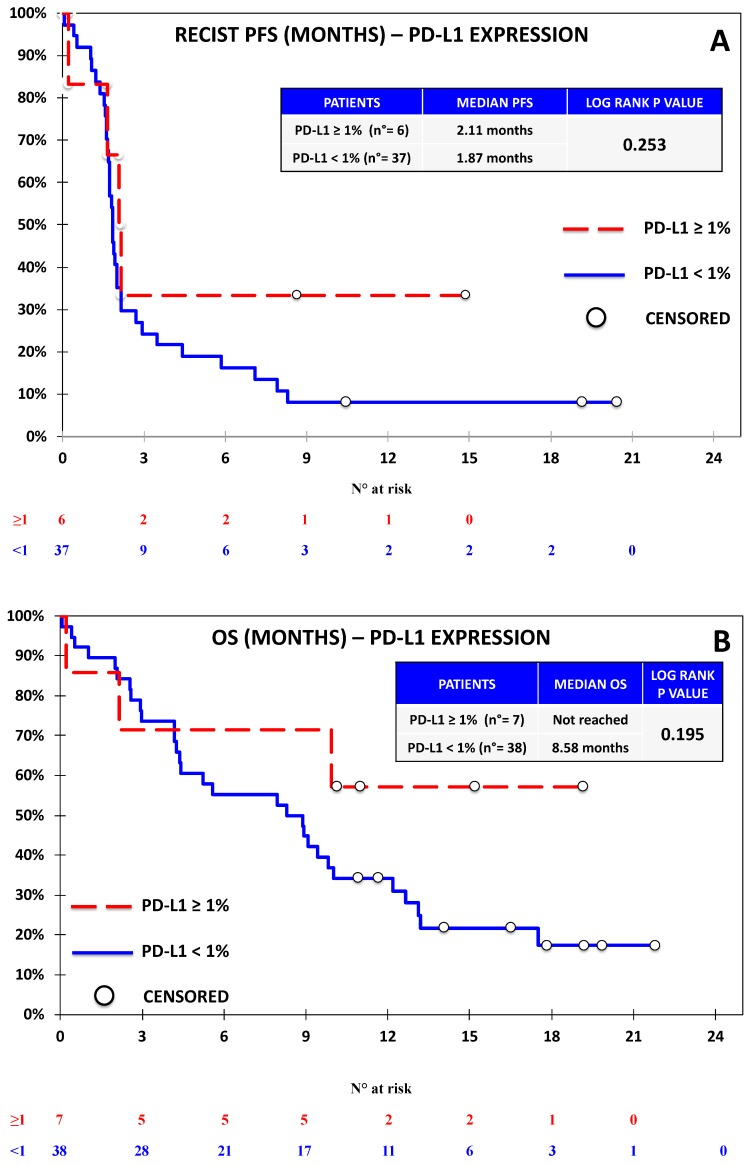
Kaplan–Meier curve for Response Evaluation Criteria in Solid Tumors (RECIST)-progression-free survival (PFS) (**A**) and overall survival (OS) (**B**) based on the expression of PD-L1 defined as ≥1% vs. <1% in the Nivolumab Cohort. No significant difference was observed on the basis of PD-L1 expression.

**Figure 3 jcm-08-01566-f003:**
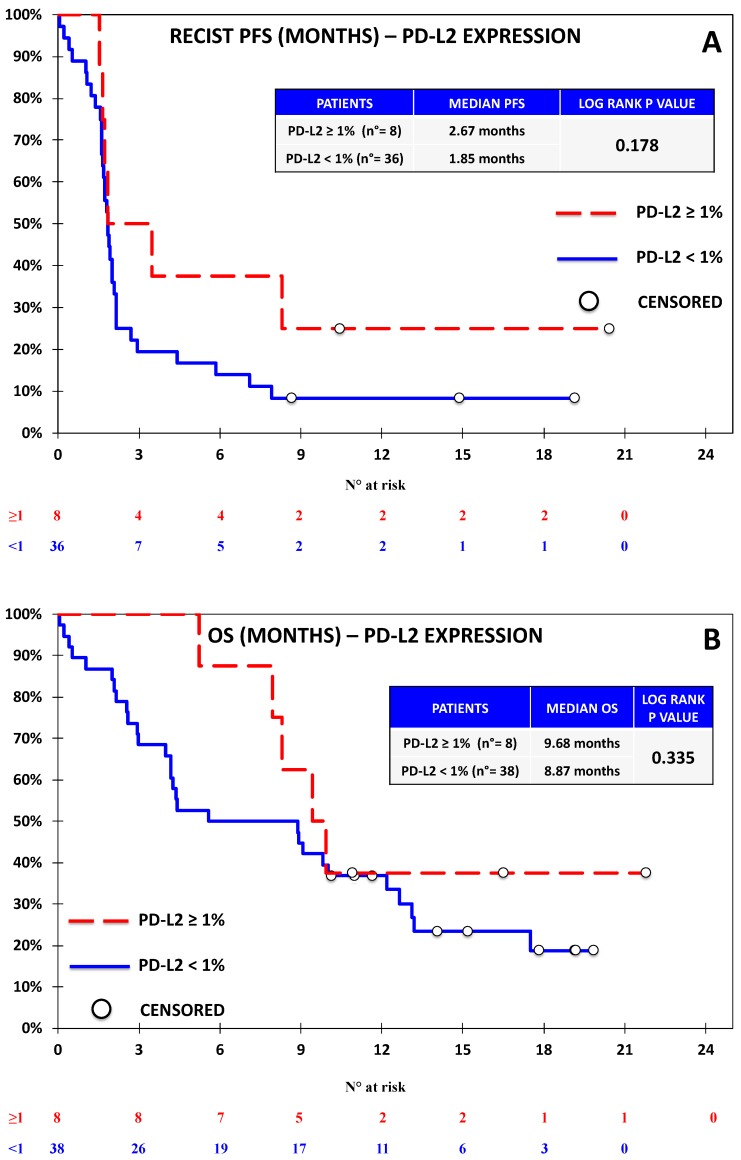
Kaplan–Meier curve for RECIST-PFS (**A**) and OS (**B**) based on the expression of PD-L2 defined as ≥1% vs. <1% in the Nivolumab Cohort. No significant difference was observed on the basis of PD-L2 expression.

**Figure 4 jcm-08-01566-f004:**
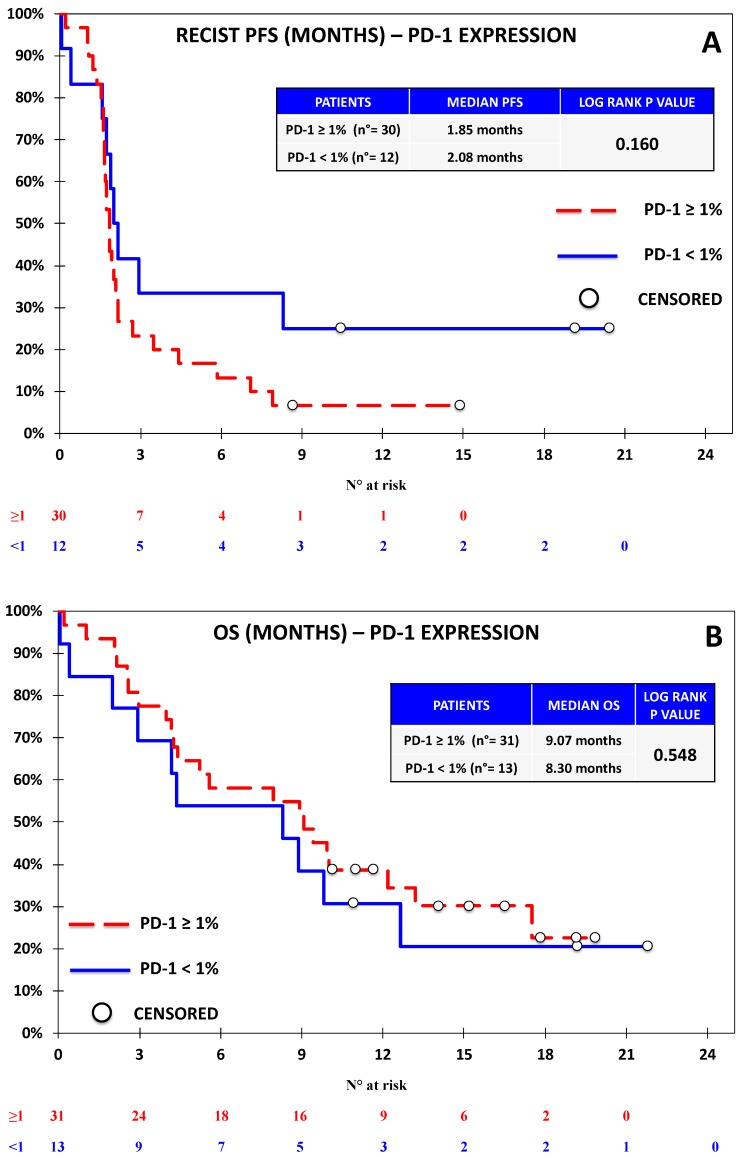
Kaplan–Meier curve for RECIST-PFS (**A**) and OS (**B**) based on the expression of PD-1 defined as ≥1% vs. <1% in the Nivolumab Cohort. No significant difference was observed on the basis of PD-1 expression.

**Figure 5 jcm-08-01566-f005:**
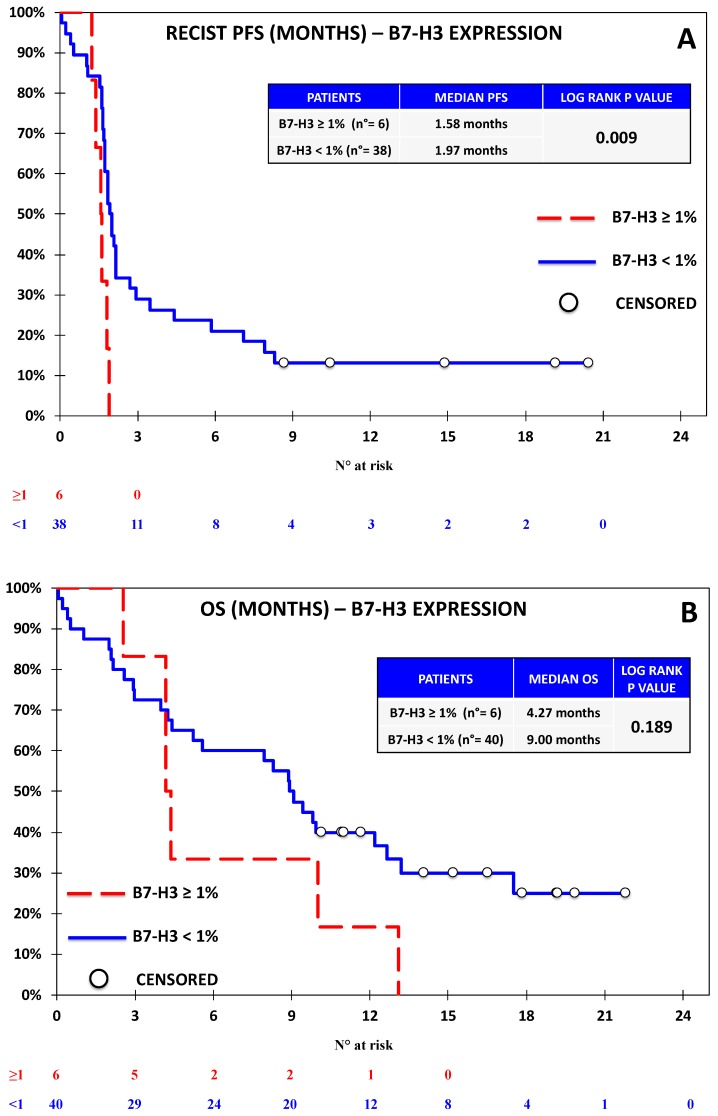
Kaplan–Meier curve for RECIST-PFS (**A**) and OS (**B**) based on the expression of B7-H3 defined as ≥1% vs. <1% in the Nivolumab Cohort. B7-H3 expression was associated with significantly lower RECIST-PFS and no significant difference in terms of OS. While these results should be considered with caution due to the fact that only six out of 44 patients expressed B7-H3, the rapid progression and death of all the B7-H3-expressing patients was noteworthy. Notably, all the patients who were alive at the time of the analysis did not express B7-H3.

**Figure 6 jcm-08-01566-f006:**
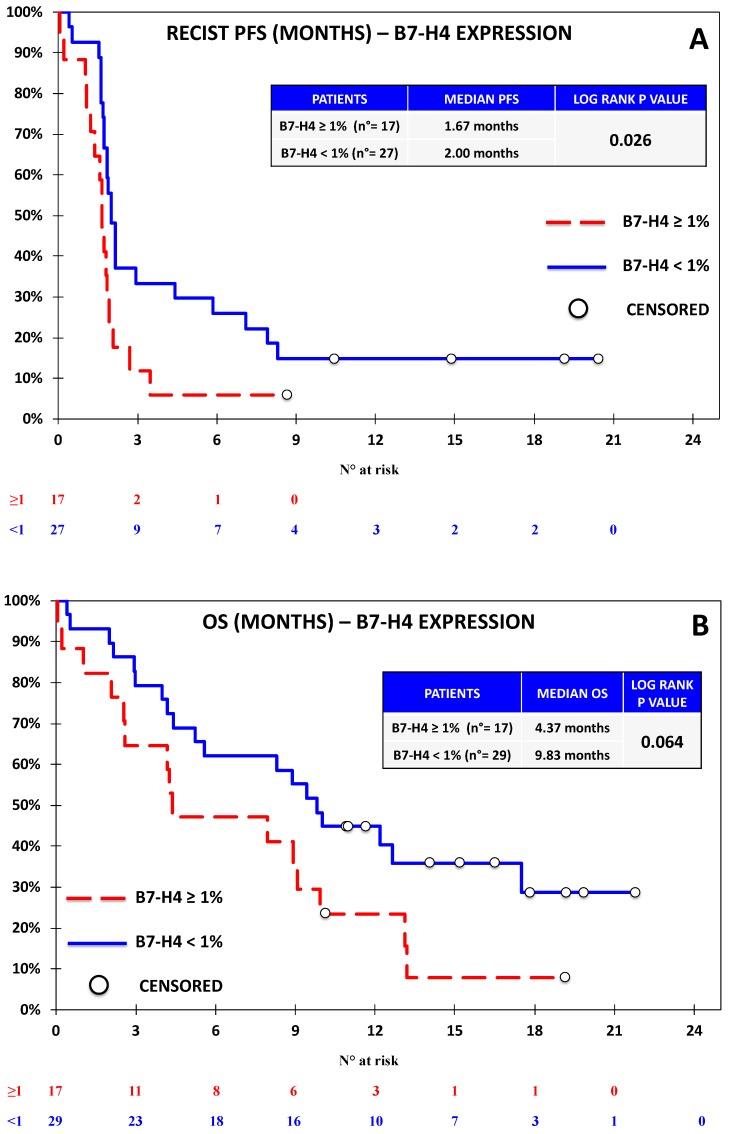
Kaplan–Meier curve for RECIST-PFS (**A**) and OS (**B**) based on the expression of B7-H4 defined as ≥1% vs. <1% in the Nivolumab Cohort. The expression of B7-H4 was significantly associated with shorter PFS and non-significantly associated with shorter OS, although the difference was close to significance (*p*-value = 0.064).

**Figure 7 jcm-08-01566-f007:**
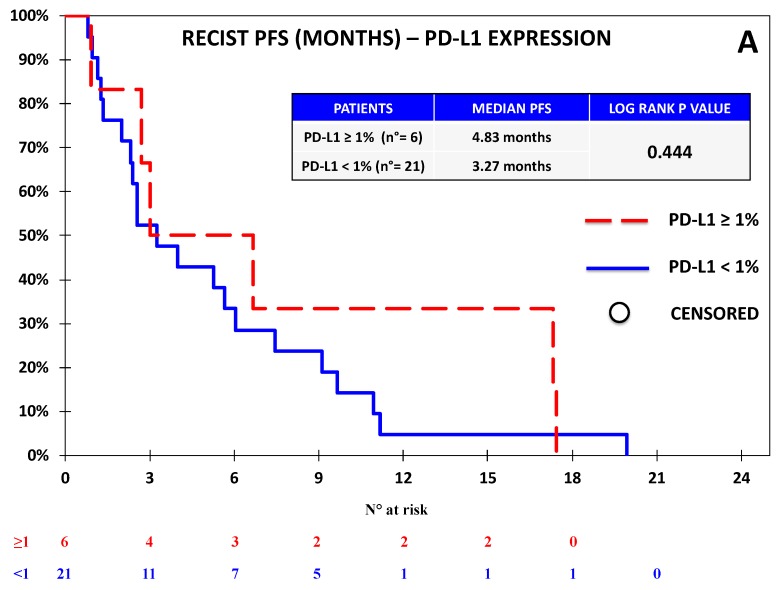
Kaplan–Meier curves for RECIST-PFS (**A**) and OS (**B**) based on the expression of PD-L1 defined as ≥1% vs. <1% in the Chemotherapy Cohort. No significant difference in terms of PFS or OS was observed on the basis of PD-L1 expression.

**Figure 8 jcm-08-01566-f008:**
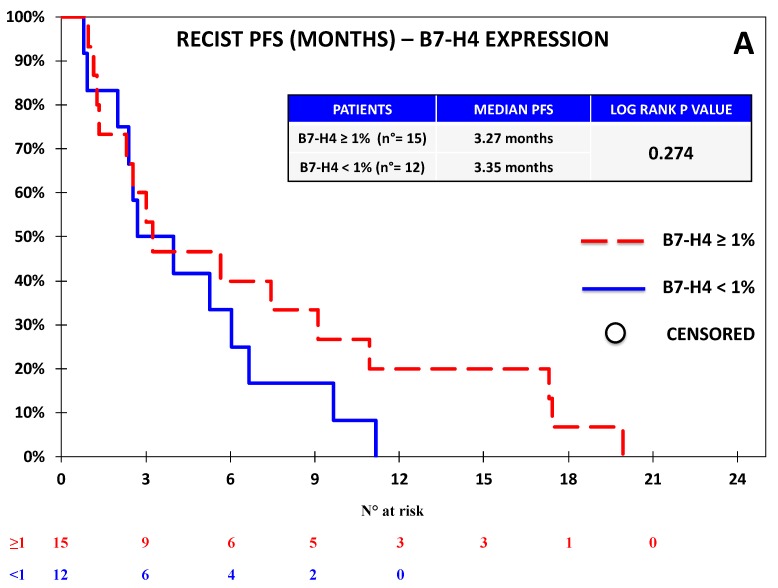
Kaplan–Meier curves for RECIST-PFS (**A**) and OS (**B**) based on the expression of B7-H4 defined as ≥1% vs. <1% in the Chemotherapy Cohort. No significant difference in terms of PFS or OS was observed on the basis of B7-H4 expression.

**Table 1 jcm-08-01566-t001:** Clinical and molecular characteristics of the evaluable patients in the Nivolumab Cohort and in the Chemotherapy Cohort. The Chemotherapy Cohort included only chemotherapy-naïve patients affected by non-squamous NSCLC with ECOG PS = 0–1.

Clinical Characteristics	Nivolumab Cohort	Chemotherapy Cohort
	*n* (%)	*n* (%)
**Patients**	46	27
**Gender**		
Male	34 (73.9)	20 (74.1)
Female	12 (26.1)	7 (25.9)
**Age (year)**		
Range	44–82	46–81
Median	70	69
**Smoking habit**		
Current	13 (28.3)	12 (44.4) ^1^
Former	24 (52.2)	12 (44.4) ^1^
Never	9 (19.5)	3 (11.1) ^1^
**Histology**		
Non-squamous	35 (76.1)	27 (100.0)
Squamous	11 (23.9)	0 (0.0)
**Stage**		
IIIB	2 (4.4)	0 (0.0)
IV	44 (95.6)	27 (100.0)
**ECOG PS**		
0	17 (37.0)	6 (22.2)
1	26 (56.5)	21 (77.8)
2	3 (6.5)	0 (0.0)
**Previous lines of treatment**		
Range	1–6	-
Median	2	-
***EGFR* mutation**		
Yes	3 (8.6) ^2,4^	1 (3.7) ^3^
No	32 (91.4) ^4^	26 (96.3)
***ALK* rearrangement**		
Yes	0 (0.0) ^4^	0 (0.0)
No	35 (100.0) ^4^	27 (100.0)

^1^ Total = 99.9% due to approximation. ^2^ Two patients in the Nivolumab Cohort had exon 19 deletion, whereas one patient had exon 19 deletion in association with exon 20 insertion. ^3^ One patient in the Chemotherapy Cohort had exon 21 mutation, which was identified after first-line treatment. ^4^ Molecular screening for epidermal growth factor receptor (EGFR) mutations and anaplastic lymphoma kinase (ALK) rearrangements in the Nivolumab Cohort was performed in non-squamous NSCLC only (35/46).

**Table 2 jcm-08-01566-t002:** Expression of the biomarkers in the two Cohorts.

	**Nivolumab Cohort *n* (%)**						
**Biomarkers**	**<1%**	**1%–9%**	**10%–49%**	**≥50%**	**ND**	**Positive Total**	**Negative Total**
**PD-L1**	38 (82.6)	4 (8.7)	2 (4.3)	1 (2.2)	1 (2.2)	7 (15.6)	38 (84.4)
**PD-L2**	38 (82.6)	2 (4.3)	5 (10.9)	1 (2.2)	0 (0.0)	8 (17.4)	38 (82.6)
**PD-1**	13 (28.3)	8 (17.4)	5 (10.9)	18 (39.1)	2 (4.3)	31 (70.5)	13 (29.5)
**B7-H3**	40 (87.0)	2 (4.3)	3 (6.5)	1 (2.2)	0 (0.0)	6 (13.0)	40 (87.0)
**B7-H4**	29 (63.0)	5 (10.9)	4 (8.7)	8 (17.4)	0 (0.0)	17 (27.0)	29 (63.0)
	**Chemotherapy Cohort *n* (%)**						
**Biomarkers**	**<1%**	**1%–9%**	**10%–49%**	**≥50%**	**ND**	**Positive Total**	**Negative Total**
**PD-L1**	21 (77.8)	5 (18.5)	1 (3.7)	0 (0.0)	0 (0.0)	6 (22.2)	21 (77.8)
**B7-H4 ^1^**	12 (44.4)	3 (11.1)	3 (11.1)	9 (33.3%)	0 (0.0%)	15 (55.6)	12 (44.4)

Expression of the potential immune-related biomarkers in the Nivolumab Cohort and in the Chemotherapy Cohort reported into each cut-off category; the evaluable samples from each cohort were then divided between positive (≥1%) and negative (<1%) expression. **^1^** Total = 99.9% due to approximation.

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
