# Peer review of "Correlation between B7-H4 and Survival of Non-Small-Cell Lung Cancer Patients Treated with Nivolumab"

_jcm, 2019, doi:10.3390/jcm8101566_

Round 1
Reviewer 1 Report
Authors answered all questions point by point, and the manuscript was extensively improved. I have no further concerns. This is an acceptable clinical study for publication.
Reviewer 2 Report
No further comments.
Reviewer 3 Report
The authors have revised their manuscript and improved the introduction and conclusion section as requested in addition to the methods and results. Hence, I do not have any objections publishing this manuscript.
This manuscript is a resubmission of an earlier submission. The following is a list of the peer review reports and author responses from that submission.
Round 1
Reviewer 1 Report
This is a very short article with numerous figures and tables. Authors attempt to demonstrate the correlation of B7-H4 with the survival of NSCLC and clarify B7-H4 is a potential predictor for the efficacy of immune checkpoint inhibitors. Actually, the similar studies in recent years are countless. Because immunotherapy is becoming the major medical option for patients with advanced cancer. To identify a powerful biomarker to predict the efficacy before therapy will be a breakthrough achievement.
Therefore, unless your study exhibits something unique, your work on immune therapy will be nothing new.
The correlation of B7-H4 with prognosis or with PD-L1/PD-1 was extensively studied. For instance:
1. Differential Expression and Significance of PD-L1, IDO-1, and B7-H4 in Human Lung Cancer.
2. Expression and clinical significance of PD-L1, B7-H3, B7-H4 and TILs in human small cell lung Cancer (SCLC)
3. B7-H3 Expression in NSCLC and Its Association with B7-H4, PD-L1 and Tumor-Infiltrating Lymphocytes
4. Association of B7-H4, PD-L1, and tumor infiltrating lymphocytes with outcomes in breast cancer
5. Characterization of immune regulatory molecules B7-H4 and PD-L1 in low and high grade endometrial tumors
Based on these findings, the current manuscript did not provide any novel concepts or new findings.
Introduction
The content is very short. The correlation of immune-related biomarkers studied by Authors was not described clearly. The rationale is also not provided well. What is the obstacle in Nivolumab-mediated therapy ?? What limitation do you want to break based on your study ?? Why your clinical findings is useful for improving tumor therapy ??
Materials
Sections of FFPE tissue were cut at 2 _m ???
To make your IHC score more reliable, H-Score or quantitated image equipment is recommended. Every specimen should have a score.
Results
The sub-title in the results shhould be re-organized. For instance:
B7-H4 correlates with ????? in Nivolumab Cohort
Something is important in Chemotherapy Cohort
Readers will easily understand what you mean before reading the detail.
Table 1 is confusion, and should be re-organized.
Figures
Authors provided numerous Figures which were not clearly described. Many important Supplementary Figures are important and should be organized in the main text. Re-organization of all Figures is required absolutely.
Figure 1
The resolution for each IHC image is poor. What is EE ?? EE should be defined.
Reviewer 2 Report
Nice research. Well-structured and written article. Relevant in the era of biomarkers.
Some remarks:
Was IHC scoring only done on tumor cells? Please specify in the Methods. Delete sentence between 3 and 3.1 headings. Please specify which of the baseline clinical markers you used for the uni- and multivariate analysis. Were these all the markers in Table 1. Some of the tables with the KM-curves are not clearly visible.
Reviewer 3 Report
In this manuscript, the authors made a correlation between different biomarkers such as PD-1, PD-L2, B7-H3, B7-H4, etc. and NSCLC patients receiving Nivolumab and found a negative predictor role of B7-H4 when compared with NSCLC patients receiving platinum-based chemotherapy. Even though the correlation was sought, and different therapy-related cohorts were described, the low quality of the figures made it difficult to fathom the underlying message.
The introduction needs to be improved and detailed to make it more inclusive. Authors can cite the following relevant paper and make the introduction more interesting: B7-H3 Expression in NSCLC and Its Association with B7-H4, PD-L1 and Tumor-Infiltrating Lymphocytes by Rimm DL.
On page:128-131, the guideline from the journal needs to be removed.
A separate conclusion needs to be added at the end of the manuscript.